# New Biomarkers of Ferric Management in Multiple Myeloma and Kidney Disease-Associated Anemia

**DOI:** 10.3390/jcm8111828

**Published:** 2019-11-01

**Authors:** Małgorzata Banaszkiewicz, Jolanta Małyszko, David H. Vesole, Karolina Woziwodzka, Artur Jurczyszyn, Marcin Żórawski, Marcin Krzanowski, Jacek Małyszko, Krzysztof Batko, Marek Kuźniewski, Katarzyna Krzanowska

**Affiliations:** 1Departament of Nephrology, Jagiellonian University Medical College, Kopernika 15-15c, 31-501 Cracow, Poland; mbanaszkiewicz@gmail.com (M.B.); woziwodzka.karolina@gmail.com (K.W.); mkrzanowski@op.pl (M.K.); batko.krzysztof@gmail.com (K.B.); marek.kuzniewski@uj.edu.pl (M.K.); 2Department of Nephrology, Dialysis and Internal Medicine, Warsaw Medical University, Banacha 1a, 02-097 Warsaw, Poland; jolmal@poczta.onet.pl; 3John Theurer Cancer Center, Hackensack University Medical Center, 92 2nd St, Hackensack, NJ 07601, USA; dvesole@yahoo.com; 4Departament of Hematology, Jagiellonian University Medical College, Kopernika 17, 30-501 Cracow, Poland; mmjurczy@cyf-kr.pl; 5Departament of Clinical Medicine, Medical University, Szpitalna 37, 15-254 Bialystok, Poland; mzorawski@wp.pl; 6Departament of Nephrology, Medical University, Żurawia 14, 15-540 Bialystok, Poland; jacek.malyszko@umb.edu.pl

**Keywords:** anaemia, growth differentiation factor 15, hepcidin, kidney disease, mieloma multiple, soluble transferrin receptor, zonulin

## Abstract

Multiple myeloma (MM) is a malignancy of clonal plasma cells accounting for approximately 10% of haematological malignancies. MM mainly affects older patients, more often males and is more frequently seen in African Americans. The most frequent manifestations of MM are anaemia, osteolytic bone lesions, kidney failure and hypercalcemia. The anaemia develops secondary to suppression of erythropoiesis by cytokine networks, similarly to the mechanism of anaemia of chronic disease. The concomitant presence of kidney failure, especially chronic kidney disease (CKD) and MM per se, leading to anaemia of chronic disease (ACD) in combination, provoked us to pose the question about their reciprocal dependence and relationship with specific biomarkers; namely, soluble transferrin receptor (sTfR), growth differentiation factor 15 (GDF15), hepcidin 25 and zonulin. One or more of these are new biomarkers of ferric management may be utilized in the near future as prognostic predictors for patients with MM and kidney failure.

## 1. Introduction

Multiple myeloma (MM) is a malignancy of clonal plasma cells accounting for 10% of haematological malignancies and 1.8% of all malignancies [1]. Because of complicated pathogenesis and multiple end-organ effects, patients with MM are treated by specialists from many areas of medicine, including haematology, nephrology, cardiology and orthopaedics. In Poland, the number of newly diagnosed patients with MM is approximately 1500–1800 per year [2]. Despite being a disease of an older population (median age 70 years old), approximately 10% of patients are under 50 years old [3,4]. Interestingly, lately, a slight increase of morbidity in 45–64 year old patients was observed [5]. With the development of novel target therapies, the projected median life expectancy of a newly diagnosed MM patient with standard risk cytogenetic features has improved from 2.5 years to over 10 years [2].

The most frequent manifestations of MM are: anaemia (73%), bone pain and osteolytic bone lesions (58% and 67%, respectively), kidney failure (48%) and hypercalcemia (28%) (CRAB) [3]. The anaemia develops secondarily to the suppression of erythropoiesis by cytokine networks, similarly to the mechanism of anaemia of chronic disease (ACD). In the majority of MM patients, the anaemia is manifested as a haemoglobin level between 8 and 10 g/dL, while approximately 10% have a value below 8 g/dL [6]. Anaemia has a negative impact on the quality of patients’ lives and is also an independent predictor of poor survival. Although disease progression exacerbates the extent of the anaemia, the use of treatment reduces the extent of bone marrow plasmacytosis and allows for improvement of haematological and renal parameters [3].

Increased use of iron in cancer cells may be responsible for findings of anaemia in the course of MM, even without coexisting kidney impairment. Iron circulation in the human body is associated with four cell types: enterocytes involved in iron absorption; erythroblasts—the precursors of red blood cells (RBCs); splenic macrophages degrading aged RBCs and releasing iron back to the bone marrow via transferrin; and hepatocytes playing a role in monitoring transferrin saturation, hepatocellular iron content, regulating iron absorption from the gut and regulating iron release from the spleen [6]. Recent scientific research has shown that disorders of iron metabolism, and consequently, anaemia, may be associated with a positive regulation of hepcidin 25 expression caused by cytokines [7]. This small peptide hormone composed of 25 amino acids synthesized in hepatocytes regulates iron absorption from the gut and iron release from the spleen [8]. Growth differentiation factor 15 (GDF15) is a member of the transforming growth factor-beta family, aberrantly secreted by bone marrow stromal cells (BMSCs) in MM and plays a role in regulation of hepcidin expression [9,10]. Soluble transferrin receptor (sTfR) reflects the functional iron compartment in serum, an increased number of erythropoietic precursor cells and an increasing iron need for erythropoiesis [11]. Zonulin is a protein, which modulates intercellular tight junctions related to the intestinal permeability of iron as well as autoimmune, inflammatory and neoplastic substances also expressed beyond the gastrointestinal tract [12]. Thus GDF15, sTfR, hepcidin 25 and zonulin may all be involved in iron metabolism, so may form new biomarkers of ferric management and one or more may ultimately serve as a predictor for MM-associated kidney failure. If one or more of these molecules are proven to be a predictive biomarker for the development of acute/chronic kidney disease, this may lead to therapeutic interventions to prevent renal insufficiency. The pathophysiology of iron metabolism is shown in Figure 1.

## 2. Pathogenesis of Anaemia in Multiple Myeloma

Anaemia is among the most frequent end-organ sequelae in MM [3]. It is an early and multifactorial complication of MM, which has been reported to occur in over two thirds of all patients [13]. Major pathophysiological mechanisms of MM-related anaemia are the underlying basis of plasma cell-produced cytokines that mark anaemia of chronic disease (ACD), which inhibit erythropoiesis and impair iron homeostasis [13]. The anaemia is usually normocytic and normochromic but can be macrocytic. Serum iron levels are usually normal to mildly low; serum ferritin is hig;h and hemosiderin is significant in bone marrow macrophages, consistent with patterns similar to ACD [14]. Relative erythropoietin deficiency, renal impairment and myelosuppressive consequences of chemotherapy are other factors to account for [13]. Recent studies continue to add to the complexity of anaemia pathophysiology in MM. Studies have shown that malignant plasma cells exhibited high up-regulation of apoptogenic receptors, which lead to immature erythroblast apoptosis, which implies that cytotoxic characteristics of myeloma cells may be crucial in provoking inadequate erythropoiesis [15]. Outside of the concept of bone marrow “crowding out” by malignant cells, the tumour microenvironment also seems to functionally impair hematopoietic stem and progenitor cells, in part through TGF-beta signalling [16]. It seems that assessment of anaemia and devising of a strategy for its treatment may be more accurately undertaken through a combination of molecular and biochemical parameters of tumour burden, cytotoxicity and the myeloma-promoting bone marrow microenvironment.

The most common causes of MM-related anaemia are: displaced erythroid system by neoplastic plasmacytes, proinflammatory activity of cytokines, disabled apoptosis of the erythroid system, inadequate excretion of erythropoietin (EPO) compared to the degree of anaemia, reduction of erythrocytes’ survival time (<10%), inadequacy of ferric management and direct suppression of erythropoiesis by neoplastic cells [17]. The last four causes are considered to be the main processes responsible for ACD development. Prospective survey data indicate that clinically, anaemia is associated with fatigue, dyspnoea, cardiovascular complications and cognitive dysfunction [18]. In addition, anaemia may be associated with permanent organ dysfunction or intensifying hypoxia, which may also lead to changes in neoplastic metabolism. This may subsequently contribute to refractoriness of chemotherapy and radiotherapy [17,18]. Low initial haemoglobin level (<14.0 g/dL for men; <12.0 g/dL for women), chemotherapy persistent/recurrent disease and female gender comprise risk factors for the development of anaemia in MM patients [18]. A diagnostic and clinical scheme of MM-related anaemia is shown in Figure 2. 

Chemotherapy-responsive patients, with or without exogenous erythropoietin administration, often see a normalization of their haemoglobin levels with a resultant improvement in quality of life (QOL), despite the impact of patients’ ages and stages of disease. It is important to acknowledge that the initiation of treatment, particularly with alkylating agents, may transiently lead to an early decrease of haemoglobin levels, followed by an increase in haemoglobin with disease response, often within the first month of therapy. This initial haemoglobin reduction is not observed during non-alkylating treatment [19].

Furthermore, treatment of anaemia in MM patients includes red blood cell (RBC) transfusions, recombinant human erythropoietin (rHuEPO) therapy and oral or parenteral iron supplementation [13]. RBC transfusions cause an immediate effect and rapid increase in haemoglobin levels. Transfusion may rarely be associated with several risks factors, including infections, mild to even life-threatening immunologic reactions, iron overload, and in rare cases, induction of graft-versus-host disease (GVHD). Transfusions are usually reserved for patients who are severely anaemic and symptomatic (haemoglobin < 7.0 g/dL) and those patients who fail to respond to chemotherapy and to erythropoietic-stimulating agents (ESAs) [17]. rHuEPO is an erythropoiesis stimulating agent (ESAs), a biological equivalent to the human endogenous hormone EPO. The application of rHuEPO leads to an increase of haemoglobin level over minimal normal concentration without the necessity of RBC transfusions [13]. ESAs are optimally incorporated with concurrent chemotherapy [13] The treatment should be considered for haemoglobin level < 10 g/L or clinical anaemia symptoms. ESAs can increase haemoglobin levels up to 2 g/dL or more in 60–75% of MM patients. Similar guidelines should be followed for the longer acting darbepoetin [20]. A benefit of improving haemoglobin levels is the subsequent improvement in overall survival. ESAs are safe and well-tolerated with minimal toxicity risk; however, pure red cell aplasia is a rare event [21]. According to the solid tumour literature, ESAs have a negative influence on survival in patients with active forms of solid tumours. Furthermore, EPO treatment enhances the circulating levels of angiogenic cytokines, contributing to the progression of neovascularization in tumours, and as a consequence, increased neoplasm growth [22]. Thus, the risk of higher mortality associated with ESAs is predominantly observed in patients with solid tumours and is a rare complication in hematologic malignancies. The most serious side effect of this treatment is a higher risk of thrombo-embolic complications, especially in patients with MM treated with immunomodulatory agents in combination with corticosteroids (e.g., thalidomide, lenalidomide and pomalidomide). Hence, it is important to consider anti-thrombotic prophylaxis (low molecular weight heparin, factor Xa inhibitors or acetylsalicylic acid) therapy in individuals with these two anti-myeloma agents [17]. For those patients who are iron deficient, appropriate work up for blood loss should be initiated. Oral iron supplementation, at least 65 mg of elemental iron daily, should be utilized. If there is no increase in haemoglobin by at least 1 g/dL within 4–6 weeks of oral iron administration, parenteral iron should be considered. There are various formulations and schedules for parenteral iron administration. For patients with moderately-severe iron deficiency, concomitant iron supplementation and ESAs should be utilized.

Moreover, sotatercept and luspatercept are recombinant, soluble activin type-II receptor–IgG-Fc fusion proteins, which may become of new form of treatment in anaemic patients. The Medalist Trial Fenaux et al. [23] examined 229 anaemic patients with very low, low, or intermediate-risk myelodysplastic syndromes (MDS) with ring sideroblasts (RS) who required RBC transfusions and observed that treatment with luspatercept resulted in a significantly reduced transfusion burden compared with placebo in patients. Two studies (NCT01146574) and (NCT01999582) have examined treatment with sotatercept in anaemic patients with end-stage renal disease on haemodialysis; however, no study results have been posted [24].

Renal failure contributes to anaemia in MM patients. It is well known that serum EPO levels below the lower limit of normal are more frequent in the population of patients with renal impairment (60%) than in the population of all patients with MM (25%). However, renal failure does not significantly affect the response to ESA therapy [21]. Soleymanian et al. [25] showed that 88% of patients with MM and renal insufficiency had anaemia, in comparison to only 73% of patients in the whole MM population [3] (Table 1). Together, these findings illustrate the importance of renal organ function in the severity of MM complications. Interestingly, Liu et al. described a group of 161 MM patients and confirmed that severe anaemia is an independent risk factor of renal impairment in this population [26]. Nevertheless, anaemia is not a useful tool in the diagnostic process for MM in patients with chronic kidney disease (CKD) because of its presence in both clinical scenarios.

## 3. Hepcidin 25

Hepcidin is a 25 amino acid, negative, iron-regulating peptide hormone produced in the liver. It controls iron delivery to the blood from intestinal cells, regulates its transport from iron-storing hepatocytes, releases it from macrophages and facilitates its transport via the placenta. Production of hepcidin is stimulated by increased plasma iron, and altered by iron stores and proinflammatory cytokines; though increased EPO activity, it has a suppressive role [17]. The mechanism of hepcidin activity depends on binding and inactivating ferroportin (described as the solitary cellular iron exporter), which inhibits iron delivery to plasma from all iron-transporting cells [27]. This hormone may play a major role in anaemia associated with chronic disease and inflammation, because of its unique regulatory role on ferric management [28].

Interleukin-6 (IL-6), is increased in MM patients and is reported as the main cytokine regulating hepcidin expression [6]. However, high hepcidin expression may not only be due to IL-6 activity, as other cytokines may also contribute to anaemia in ACD. Ibricevic-Balic et al. [8] reported that increased serum hepcidin concentration may lead to anaemia in MM. In their study, patients with newly diagnosed MM (27 patients) and healthy controls (60 people) were examined. Anaemia was observed in 70% cases of MM patients. However, despite significantly elevated serum hepcidin concentration in the anaemic MM group, no correlation between hepcidin and IL-6 was found. Similar results were observed by Sharma et al. [29], who demonstrated a strong association between up-regulated hepcidin expression and anaemia in advanced stage MM. In their study, the correlation between hepcidin concentrations and serum IL-6 levels showed borderline significance. Moreover, strong correlation between urinary hepcidin with serum ferritin and C-reactive protein (CRP) was observed, confirming hepcidin expression as a part of an acute-phase reaction. They did not observe an association between other proinflammatory cytokines (i.e., serum tumour necrosis factor-α (TNF-α) or interleukin-1 β (IL-1β)) and hepcidin. The lack of consistent associations between Il-6 (or other cytokines) and hepcidin indicates that other stimuli of hepcidin regulation (e.g., pathways associated with iron storage) may be deciding factors. Victor et al. [30] observed a strong negative correlation between serum hepcidin and inflammatory markers (i.e., IL-6 and CRP) in a group of 21 newly diagnosed MM patients. In contrast, Han et al. [31,32] demonstrated that serum IL-6 positively correlated with monocyte hepcidin [31]. Moreover, the expression level of monocyte hepcidin mRNA positively correlated with serum ferritin and IL-6 levels, but was unrelated with TNF-α level [31] in untreated patients. Mei et al. [33] observed a significant positive correlation between plasma IL-6 level and hepcidin mRNA expression. Together, those studies illustrate the complex relationship between hepcidin and IL-6 in the development of anaemia in MM.

According to Maes et al. [14] bone morphogenetic protein 2 (BMP-2) is a major mediator of the hepcidin stimulatory activity in MM. They reported data on 25 MM patients and found increased BMP-2 levels. It was discussed that IL-6 and BMP may stimulate hepcidin promoter activity in a synergistic manner with postulated crosstalk between the two signalling pathways. Hence, the presence of IL-6 and BMP2 together in the serum may act jointly and lead to increased hepcidin expression.

Katodritou et al. [34] observed that effective MM treatment decreased the abnormally high serum hepcidin levels with subsequent improvement in anaemia. Among 34 anaemic MM patients treated using immunomodulatory drug-based therapies (thalidomide or lenalidomide combinations with cyclophosphamide or dexamethasone) or conventional therapy, 80% responded to MM treatment. In responders, elevated serum hepcidin levels significantly decreased during therapy. Moreover, hepcidin serum levels predicted the reduction of anaemia in response to therapy around the first month. A Lower hepcidin level was associated with improvement of anaemia. Similar results were reported by Mei et al. [33] in 25 MM patients with significantly decreased levels of plasma hepcidin, who achieved complete remission after six cycles of bortezomib and dexamethasone chemotherapy. It should be noted that these studies have small sample sizes, which may be considerably affected by individual disease characteristics and the underlying treatment regimen. However, it can be observed that an optimal treatment outcome, which likely reflects a reduction in tumour load, leads to improvement, and potential resolution of anaemia.

Interestingly, Haraguchi et al. [35] showed that MM patients with renal insufficiency had significantly higher pro-hepcidin (hepcidin prohormone) levels compared to patients with appropriate renal function or impairment. However, no correlation between pro-hepcidin levels and serum iron, ferritin or haemoglobin concentrations were observed. Since hepcidin is cleared through the kidney route, urinary hepcidin levels may not correctly reflect the serum levels in renal disease because of changes in filtration via glomerular membrane or reabsorption and degradation in the proximal tubules. However, assessing pro-hormone levels may be limited in the interpretation of the biological activity of hepcidin. Łukaszyk et al. [36] reported on 69 patients with early stages of CKD, and showed higher serum hepcidin levels in patients with functional iron deficiency, when comparing to patients with absolute iron deficiency. They observed that hepcidin was predicted by circulating markers of inflammation, such as ferritin, fibrinogen and IL-6.

An overview of the characteristics of hepcidin is displayed in Table 2, with a focus on its biomarker potential, and factors that may influence its adequate assessment. Studies described in this section point to the difficulties of the adequate interpretation of hepcidin concentrations, which require an understanding of the clinical scenario (i.e., state of hematologic disease, iron deficiency status and kidney function) (direction of changes described in Table 3). Interplay between circulating inflammatory molecules and pathways signalling the status of iron stores may exert stimuli of variable degrees, to which the response in hepcidin regulation is not fully known. Overall, a response to myeloma treatment, and reduced tumour burden, seem to be paralleled by an improvement of anaemia, which could be predicted with hepcidin assessments. However, difficulties in the accurate assessment of hepcidin require studies with uniform methodology to determine the clinical feasibility of this macromolecule [28,37].

## 4. Growth Differentiation Factor 15 (GDF15)

Growth differentiation factor 15 (GDF15) is considered a member of transforming growth factor beta (TGF-beta) superfamily, though it was also discovered to be highly expressed in erythroblasts, and therefore may play a major role in regulation of the erythroid lineage [39]. Multiple roles for inflammatory modulation (i.e., in an inhibitory capacity) and tumorigenesis (both pro- and antitumoral activity) have been proposed for GDF15, though its biological activity has not been elucidated [39]. GDF-15 has been associated with poor treatment response in MM, though survival analyses did not show an effect, possibly due to a small sample size of the investigation [45]. The ability GDF15 to contribute to anaemia’s development may follow from its regulation of hepcidin expression. According to Tanno et al. [10], BMSCs produce GDF15 after direct contact with plasma cells. Abnormal secretion of GDF15 is observed from bone marrow stromal cells (BMSCs), and studies have focused on the unique role of the myeloma microenvironment in promoting clonogenic growth and self-renewal via GDF15 [10]. It has been noticed that aside from “overcrowding,” there is functional impairment in hematopoietic stem and progenitor cells of myeloma, which is tied to TGF-β signalling [16]. Preliminary studies have indicated the clinical potential of measuring GDF15 to predict disease progression and outcomes [10,41], which may also indirectly influence an assessment of anaemia, as reducing bone marrow infiltration by malignant cells should improve haematopoiesis.

Mei et al. [33] showed that the levels of GDF15 were significantly higher in MM patients than among healthy controls. In addition, MM patients with higher stage disease had substantially higher GDF15 levels. Moreover, the concentrations of GDF15 were significantly decreased in MM patients in complete remission, which implies the deleterious role of tumour burden in potentiating the development of end-organ complications. No correlation between the expression of GDF15 mRNA and serum ferritin level was observed, which may suggest that an interplay occurs downstream. It is valuable to consider the complementary findings of Tarkun et al. [45], who observed significantly higher levels of GDF15 in newly diagnosed MM patients (*n* = 35), and negative correlations between GDF15 and haemoglobin. Corre et al. reported similar findings, and the personal research of this author has also indicated that recombinant GDF15 had a slightly-inhibitory in vitro effect on haematopoiesis [39,41]. These results are also consistent with those reported by Westhrin et al. [9], who described elevated serum concentrations of GDF15 in MM patients compared to controls. An increase in serum GDF15 levels by 1 ng/mL was associated with an increased risk of death of 1.187. Moreover, GDF15 was associated with osteolytic bone disease, which may suggest that GDF15 is a close surrogate of end-organ effects of MM [9]. Indeed, Windrichova et al. observed that GDF15 may be a new biomarker used in monitoring bone metastatic disease [46].

Corre et al. [41] reported on 131 MM patients, observing that GDF15 is associated with survival and may be a potential factor of treatment-refractory MM cells; for example, GDF15-induced resistance mainly to melphalan and bortezomib in stroma-dependent and stroma-independent MM cells. There was a lower probability of event-free and overall survival in patients with high levels of plasma GDF15. This data suggests that GDF15 may function as predictor of patient survival in MM.

Łukaszyk et al. [36] observed that patients with early stages of CKD and functional iron deficiency had higher GDF15 serum levels in comparison to patients without iron deficiency. Zhao et al. [47] examined 24 pre-treatment patients with MM and observed a positive correlation between serum GDF15 level and serum creatinine. These results are consistent with the findings of Tarkun et al. [45], who reported a strong association between serum GDF15 level and creatinine. Lukaszyk et al. [48] examined variations of serum GDF15 level in 87 patients with early stages of chronic kidney disease (CKD). According to this study, patients ≥ 65 years old and with anaemic status had increased serum concentrations of GDF15, which in itself correlated with haemoglobin and glomerular filtration. Moreover, haemoglobin was described as a potential predictor of GDF15.

According to Mei et al. [33], concentrations of serum hepcidin and GDF15 are similar in patients with MM. This study revealed significantly elevated serum hepcidin and GDF15 levels not only in MM patients as a whole, but also in subgroups depending on disease stage. Both serum hepcidin and GDF15 levels were significantly decreased in MM patients after effective treatment. However, direct association between these two biomarkers was not analysed. These findings lead us to propose that both GDF15 and hepcidin have to be considered with the primary disease in mind; our understanding extrapolated from other diseases, even anaemia of chronic disease, may lack consideration of stimuli originating from a malignant plasma clone.

## 5. Soluble Transferrin Receptor (sTfR)

Iron, despite being an essential element for life, can be also toxic by its influence on oxygen radical production. The body’s protective mechanism to prevent iron-associated toxicity is found in transferrin, an iron transport protein in blood, and transferrin receptor (TfR) binding to iron-loaded transferrin and internalizing it by endocytosis. Of two TfR types, TfR1 is more important in anaemia associated with MM. TfR1 has higher affinity for transferrin and is overexpressed on cells with a high rate of proliferation, including malignant hematopoietic neoplasms; conversely, TfR2 is notably restricted to hepatocytes [6]. Nevertheless, Takubo et al. [49] reported that in a study of 82 patients with hematologic malignancies, significantly higher serum sTfR levels were not observed in MM, in contrast to other examined diseases, such as acute non-lymphocytic leukaemia, chronic myeloproliferative disorders, myelodysplastic syndrome and lymphomas.

In theory, sTfR measurement may be the most important biomarker to explain the etiology of anaemia in MM compared to more established biomarkers, such as serum ferritin, transferrin saturation, ferritin index (sTfR/log transferrin saturation), hypochromic reticulocytes and C-reactive protein (CRP). Using sTfR alone in the differential diagnosis of anaemia and the prediction of treatment response is of low predictive value. Using the sTfR/log value in ferritin index may help differentiate functional iron deficiency and ACD [50]. It is accepted that serum ferritin represents the iron storage compartment, and sTfR, the functional iron compartment. Katodritou et al. [11] reinforce this hypothesis in their study of 26 patients with newly diagnosed and anaemic MM patients (haemoglobin ≤ 10.5g/dL): the combination of the ferritin index and hypochromic erythrocytes was predictive. This index may delineate the patients who will benefit from recombinant human EPO and identify functional iron deficiency requiring iron supplementation during diagnosis and early treatment stages. Łukaszyk et al. [36] proved elevated sTfR serum levels in patients with functional iron deficiency. In that group, sTfR did not correlate with any of the inflammatory parameters. On the other hand, a strong correlation with renal function parameters in absolute iron deficiency was observed. Yin et al. [51] showed that sTfR serum levels can be used as a marker of erythropoiesis in CKD patients treated by high-flux haemodialysis.

Kostova et al. [52] examined 42 patients with MM and observed an inverse correlation between EPO and haemoglobin in patients with MM and preserved renal function, in contrast to those with renal insufficiency. Interestingly, 43% of patients without renal insufficiency (RI) and 85% of patients with renal insufficiency had inadequate EPO responses to anaemia. Inadequate sTfR response to anaemia was found in 76% of all patients. In both groups with and without renal insufficiency, a positive correlation between haemoglobin and sTfR was demonstrated. Alam et al. [53] reported on 140 patients with CKD and 44 healthy controls showing increased sTfR serum levels in line with CKD stages. Similarly, ferritin and sTfR to ferritin ratio was decreased in advanced stages. This supports the concept that joint assessment of ferritin and sTfR is a valuable index of renal function in CKD.

## 6. Zonulin

Zonulin is a eukaryotic equivalent of the Vibrio cholerae zonula occludens toxin, which may reversibly regulate the intestinal permeability through altering intercellular tight junctions’ function. Proteomic studies in human sera have shown that zonulin matches to the precursor of haptoglobin-2, while its ability to alter intestinal permeability seems to occur through transactivation of the epidermal growth factor receptor (EGFR) by its single-chain form [54]. The bodily function of haptoglobin is still not fully elucidated, though it seems to play a role in preventing oxidative injury by binding with haemoglobin [55]. Zonulin, in its cleaved, double-chain form has been characterized as a haemoglobin scavenger [54]. Studies report widespread expression of zonulin in the gastroesophageal mucosa [56]. Zonulin has also been implicated in diseases associated with altered intestinal barrier function, where it may play a role in tolerance and the immune response to antigens [57]. In the study of Kume et al., 80 obese children showed elevated serum zonulin levels [58]. A negative correlation of zonulin with HDL-C, and a positive one with leptin levels, were observed, even after adjustments for age and body mass index. Moreover, Ohlsson et al. studied a population of 363 subjects, demonstrating higher zonulin to be associated with higher waist circumference, diastolic blood pressure, fasting glucose and elevated risk of metabolic disorders [59]. In addition, serum concentrations of zonulin have been tied to inflammatory indices (tumour necrosis factor alpha and interleukin 6) and age, being reportedly higher in older (≥70 years old) subjects [60]. Data on the dietary impacts on zonulin levels are inconclusive, to date [59]. 

Dschietzig et al. observed an association between serum zonulin levels and kidney failure [61]. In their study on 225 patients carrying automatic implantable cardioverters/defibrillators (AICD) for primary or secondary prevention, an inverse association was demonstrated between zonulin and serum creatinine levels. In the same study, examining a group of patients with diastolic or systolic chronic heart failure, no association was observed between zonulin levels and chronic heart failure severity [61]. Malyszko et al. aimed to establish the relationship between zonulin, iron status and anaemia in a study of kidney allograft recipients and healthy controls, observing that zonulin was substantially lower in the former group; was correlated with erythrocyte counts, haemoglobin and the haematocrit; but was not with iron parameters [55]. The authors discussed the potential significance of zonulin as a marker of impaired defensive mechanism, potentially tied to systemic inflammation, or an altered response to immune-suppressing therapy. Studies in early stages of CKD, which is often described as a state of subclinical inflammation, showed no correlations between zonulin and kidney function, though associations with interleukin-6 and hemojuvelin, a protein involved in hepcidin regulation and iron overload, were observed. However, subsequent studies showed that zonulin is not related to anaemic or inflammatory status in CKD, also underscoring the variability of serum zonulin under immune responses, which limits the clinical feasibility of singular blood-drawing assays [62]. To the best of our knowledge, no study on zonulin levels has been conducted in MM, while the physiological importance of this molecule still remains to be elucidated.

## 7. Conclusions

New biomarkers of ferric management, namely, soluble transferrin receptor (sTfR), growth differentiation factor 15 (GDF15) and hepcidin 25, may correlate with characteristics in the etiology and management of MM-associated anaemia. We anticipate that new biomarkers will be incorporated into routine diagnostic and prognostic evaluations and may lead to the development of biomarker-targeted therapeutic interventions. Early intervention may result in improved disease management and an increase quality of life.

## Figures and Tables

**Figure 1 jcm-08-01828-f001:**
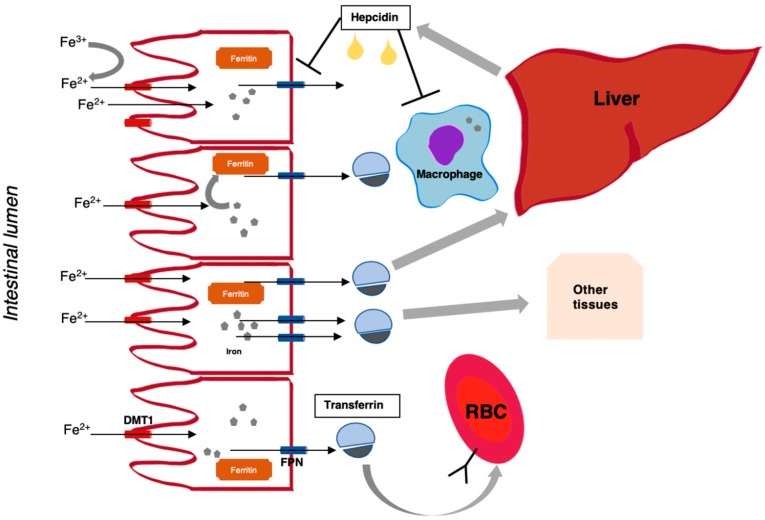
Overview of iron physiology. Non-haem iron (after reduction to Fe^2+^ by membrane-bound enzymes), originating from dietary intake (mostly Fe^3+^), enters the enterocyte of the proximal intestine via the divalent metal transporter (DMT1), and exits the cell on the basolateral side through ferroportin (FPN), which is potentiated by an oxidase, hephaestein. Depending on the state of iron need, iron can be released from the cell or sequestered in the form of ferritin (a measure of body iron stores). Iron is bound by transferrin in plasma, which enables transport and distribution to other body tissues (this also depends on the presence of transferrin receptors). Hepcidin, which is a peptide hormone produced by the liver, negatively regulates iron absorption and release from cells (e.g., macrophages) depending on iron need.

**Figure 2 jcm-08-01828-f002:**
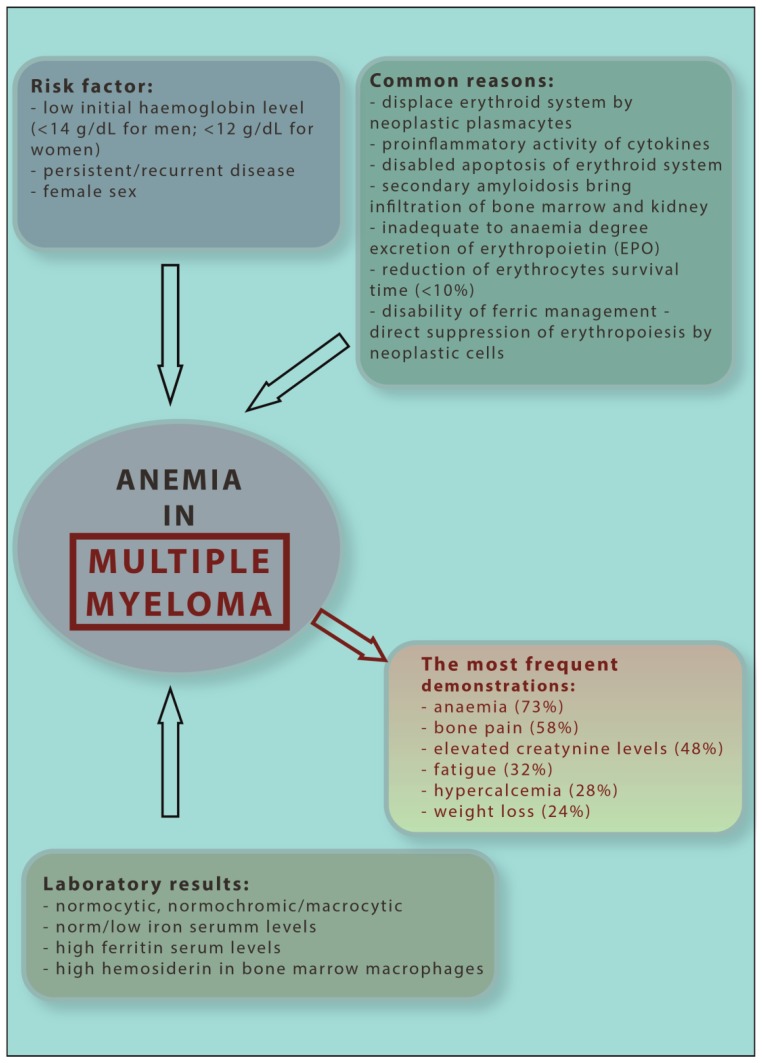
Diagnostic and clinical scheme of multiple myeloma anaemia.

**Table 1 jcm-08-01828-t001:** Anaemia in multiple myeloma (MM) and kidney failure.

	Population all Patients with MM	Population Patients with MM and Renal Insufficiency
Anaemia	73%	88%
Serum EPO levels under lower line	25%	60%

**Table 2 jcm-08-01828-t002:** New biomarkers of processes involved in myeloma-associated complications with a focus on ferric management and anaemia.

Biomarker	Sample	Characteristics
Hepcidin25 [28,37]	serum, urine	○ 25 amino acid peptide hormone sources of hepcidin include hepatocytes (major), activated neutrophils and macrophages ○ acts on ferroportin, leading to its degradation, which may restrict the iron available for erythropoiesis ○ clearance through cellular degradation and the kidney route ○ serum levels correlate well with the urinary form [37] and with ferritin, which indicates its relationship with body iron stores ○ systemic conditions (i.e., myeloma, inflammation and kidney disease), iron stores, iron intake and diurnal variability may affect hepcidin levels.
GDF15 [38,39,40,41]	serum, urine	○ divergent member of the TGF-beta family ○ bone marrow stromal cells may be a major source in MM, while serum concentrations correlate with bone marrow levels ○ pro-inflammatory cytokines induce GDF15 expression in macrophages ○ in vitro experiments of a hepatocyte system indicate GDF-15 may reduce mRNA expression of hepcidin, and thus contribute to anaemia ○ serum concentrations may be affected by kidney function, and iron deficiency described as an “early response molecule to tissue injury” in the setting of cardiovascular disease, and shown to predict decline in glomerular filtration ○ currently investigated in a wide range of diseases (neoplasms, cardiovascular, renal) ○ prognostic role in AL amyloidosis recently reported
**sTfR** [42,43,44]	serum, urine	○ single polypeptide chain ○ reported to parallel total body mass of cellular TfR ○ large studies have demonstrated normal distribution, and no relationship with age, nor gender ○ changes in iron and immune status may affect sTfR levels ○ sTfR are associated with the rate of erythropoiesis when there are adequate iron stores, irrespective of the effectiveness of the process ○ in states of iron restriction, sTfR increases; which reflects ineffective erythropoiesis and up-regulation from the deficiency of the element. Conversely, in chronic disease erythropoiesis is not enhanced and sTfR are dependent on iron supply ○ serum levels may be useful in differential of true and functional iron deficiency, assessment in inflammatory conditions and chronic disease, as well as monitoring of response to EPO ○ different assays have various methodology and reference ranges (although WHO reference reagents have been developed), which underscores a careful cross-study comparison

**Table 3 jcm-08-01828-t003:** Change in serum levels of hepcidin, GDF15, sTfR and zonulin with respect to renal disease, anaemia and multiple myeloma.

	MM	Anaemia	CKD
hepcidin 25	H	H	H
GDF15	H	H	H
sTfR	N/H	H	H
zonulin	unexamined	N	L

H—High; L—Low; N—Norm.

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
