# Peer review of "New Biomarkers of Ferric Management in Multiple Myeloma and Kidney Disease-Associated Anemia"

_jcm, 2019, doi:10.3390/jcm8111828_

Round 1

Reviewer 1 Report

In this review, authors focus on anemia of chronic disease (ACD) in MM as one of the most frequent manifestations in the myeloma pathogenesis with the reduction of hemoglobin levels and a negative impact on the quality of patients ‘life. They examine in depth four biomarkers that should be routine diagnostic factors in MM disease: soluble transferrin receptor (sTfR), growth differentiation factor 15 (GDF15), hepcidin 25 and zonulin.

Minor Remarks:

In the Introduction section authors introduce the Figure 1 that is missing of explicative legend. The authors should add it. Authors show 4 tables in the text but they don’t explain any table. They should explain and discuss them in detail in each paragraph respectively. In the text, there are typing mistakes and all layout issues should be resolved in order to submit a neat paper. The authors should carry out an important and careful improvement.

Author Response

Reviewer 1

Comments and Suggestions for Authors

In this review, authors focus on anemia of chronic disease (ACD) in MM as one of the most frequent manifestations in the myeloma pathogenesis with the reduction of hemoglobin levels and a negative impact on the quality of patients ‘life. They examine in depth four biomarkers that should be routine diagnostic factors in MM disease: soluble transferrin receptor (sTfR), growth differentiation factor 15 (GDF15), hepcidin 25 and zonulin.

We would like to thank the Reviewer for their valuable time and effort. We followed all the suggestions and have amended the manuscript.

Minor Remarks:

In the Introduction section authors introduce the Figure 1 that is missing of explicative legend. The authors should add it. Authors show 4 tables in the text but they don’t explain any table. They should explain and discuss them in detail in each paragraph respectively. In the text, there are typing mistakes and all layout issues should be resolved in order to submit a neat paper. The authors should carry out an important and careful improvement.

Thank you for your kind suggestions. We revised out manuscript extensively, since the Figures and Tables could be improved to reduce redundancy with the text itself. More concise and clear descriptions have been provided, iron physiology is now also described. Layout has been modified and typing mistakes were corrected. We reduced the number of tables, with one major table for the characteristics of biomarkers, which is now complementary to the text to prevent a repetition of information.

Reviewer 2 Report

  The manuscript is well written, and the concept proposed by the authors is interesting, but there are various defects to support their claim.

New biomarkers that you mentioned in this manuscript are only focused on ferric management. So, why don’t you change the title from “New biomarkers of anemia in multiple myeloma and kidney disease” to “New biomarkers of ferric management in multiple myeloma and kidney disease-associated anemia”?

You mentioned the mechanisms of MM-related anemia in this paper. There are more mechanisms about that, for example, abnormal up-regulation of Fas-ligand and tumor necrosis factor-related apoptosis-inducing ligand mediated cytotoxicity causes the progressive destruction of the erythroid matrix (Silvestris et al. Blood 99:1305-1313, 2002.). Furthermore, it has reported that activated TGF-beta signaling causes the functional impairment of hematopoietic stem and progenitor cells, it leads to anemia in MM (Bruns et al. Blood 120:2620-2630,2012.) It would be nice you describe them and add these references in your manuscript.

We do not need the detailed information the way to rHuEpo administration. You should remove from Line 118 to 122.

You described “in the urine” in ACD in Line 168, but in Table 3, you mentioned “in serum” levels of hepcidin 25. Which is the correct one?

This paper is focused on biomarkers of anemia in MM and kidney disease. Therefore, zonulin associated with heart transplantation seems miss the target. You should remove the sentences form Line 327 to Line 334.

   6.  The Table 4 is not easy to understand, especially the "Disadvantages” column of all biomarkers. Why most of these items correspond to “Disadvantages”? I think you do not have to arrange “the advantages and disadvantages” of new biomarkers in a table like Table 4. 

Author Response

Reviewer 2

We would like to thank the Reviewer for their valuable time and effort. We followed all the suggestions and have amended the manuscript.

  The manuscript is well written, and the concept proposed by the authors is interesting, but there are various defects to support their claim.

New biomarkers that you mentioned in this manuscript are only focused on ferric management. So, why don’t you change the title from “New biomarkers of anemia in multiple myeloma and kidney disease” to “New biomarkers of ferric management in multiple myeloma and kidney disease-associated anemia”?

 Thank you for noticing and suggesting this. Done.

You mentioned the mechanisms of MM-related anemia in this paper. There are more mechanisms about that, for example, abnormal up-regulation of Fas-ligand and tumor necrosis factor-related apoptosis-inducing ligand mediated cytotoxicity causes the progressive destruction of the erythroid matrix (Silvestris et al. Blood 99:1305-1313, 2002.). Furthermore, it has reported that activated TGF-beta signaling causes the functional impairment of hematopoietic stem and progenitor cells, it leads to anemia in MM (Bruns et al. Blood 120:2620-2630,2012.) It would be nice you describe them and add these references in your manuscript.

 Againg, thank you. We expanded this section including the recommended references.

We do not need the detailed information the way to rHuEpo administration. You should remove from Line 118 to 122.

 Done.

You described “in the urine” in ACD in Line 168, but in Table 3, you mentioned “in serum” levels of hepcidin 25. Which is the correct one?

 Measurement is possible in both serum and urine. We revised the table extensively to make this more clear.

This paper is focused on biomarkers of anemia in MM and kidney disease. Therefore, zonulin associated with heart transplantation seems miss the target. You should remove the sentences form Line 327 to Line 334.

 Done.

The Table 4 is not easy to understand, especially the "Disadvantages” column of all biomarkers. Why most of these items correspond to “Disadvantages”? I think you do not have to arrange “the advantages and disadvantages” of new biomarkers in a table like Table 4. 

Thank you for this valuable suggestion. Considering the repetitive nature of the tables with the text, we revised the tables to be more focused on biomarker potential, and less redundant.

.
